# The Impact of COVID-19 on the Italian Footwear Supply Chain of Small and Medium-Sized Enterprises (SMEs)—Evaluation of Two Case Studies

Marcello Braglia [1], Leonardo Marrazzini [1,*]  and Luca Padellini [2]

1    Department of Civil and Industrial Engineering, University of Pisa, Largo Lucio Lazzarino 2, 56126 Pisa, Italy; marcello.braglia@unipi.it
2    Department of Information Engineering, University of Pisa, Via Girolamo Caruso 16, 56122 Pisa, Italy; luca.padellini@phd.unipi.it
*    Correspondence: leonardo.marrazzini@unipi.it

**Abstract:** This paper aims to provide a case study related to two small and medium-sized enterprises (SMEs) of the Italian footwear supply chain, comparing sales and production data from pre-pandemic years with those affected by the COVID-19 (SARS-CoV-2) pandemic. Specifically, two Tuscany companies in the world of fashion footwear sector have been assisted in the analysis of their supply chains. In particular, the case research method has been employed for theory building to evaluate how companies reacted to the disruption caused by the COVID-19 pandemic to focus on potential resilience strategies that could be adopted to deal with a disruption, such as that caused by the COVID-19 pandemic. Specifically, in order to understand the dynamics of the supply chains, the standard production processes were initially studied and mapped, analyzing in detail the planning, programming, and control phase. After conducting a descriptive analysis of the data, possible resilience factors of SMEs' fashion supply chains have been extracted, and strategies that SMEs could adopt to better cope with the disruption caused by the pandemic have been suggested. The outcomes of this study can be used by decision-makers to predict the operative and long-term impacts of epidemic outbreaks on the supply chains with some suggestions of potential resilience improvement strategies. The paper is concluded by summarizing the most important insights and outlining future research agenda.

**Keywords:** supply chain management; COVID-19 pandemic; fashion industry; small and medium-sized enterprises; planning and scheduling; data analysis; resilience

## 1. Introduction

The fashion sector is characterized by short life cycles of products, high volatility, and low predictability of the demand market, linked to market trends and high impulse purchasing [1]. Short life cycles and high unpredictability of demand are usually a result of the fact that the models on sale try to capture the trends of the moment, which are highly variable and driven by phenomena such as the cinema and social media. Moreover, due to the high degree of customization of the products (e.g., color, size, style), there is a further increase of the difficulties in production management, which lead to extended intervals of time (typically about six months from the commissioning of the product to the exposition at the final customer [2,3]), leading to the transfer of semi-finished and finished products between multiple players in the supply chain (e.g., sole factories, insole factories, heel factories).

In order to improve cost efficiency, many fashion multinationals have identified Asian countries (i.e., Bangladesh, China, and India) as locations to offshore their production [4]. Unfortunately, the recent coronavirus (COVID-19/SARS-CoV-2) outbreak came from the Wuhan area, China, and immediately impacted Chinese exports, drastically reducing the supply availability in global supply chains. In Reference [5], Araz et al. underline that the

COVID-19 outbreak represents one of the major disruptions encountered during the last decades which is "breaking many global supply chains". In the period from 20 January to 5 February 2020, the number of confirmed cases of coronavirus in China rose from 292 to 28,018 cases, with a further increase to 80,880 cases as of 16 March [6]. In the last decade of February and early in March 2020, the number of COVID-19 cases has exponentially increased in Asia, Europe, and the USA, resulting in border closures and quarantine.

In such a turbulent environment, the supply chains of many companies became specifically prone to pandemic outbreaks. In addition, the fashion industry was considered non-essential. Having international suppliers had repercussions on the entire supply chain, slowing down production due to missing components. For instance, 70% of woven fabrics used in the garment industry are based in Bangladesh, and 90% of the ones based in Myanmar are sourced from China [7]. In Reference [8], Ivanov demonstrated that the lack of supply chain coordination on the timing of opening/closures negatively impacted supply chains operations and performance.

The COVID-19 pandemic led to the closure of shops, accelerated the shift towards a more digital world and triggered changes in online shopping. During the pandemic, online consumption habits have changed significantly, with a greater proportion of internet users buying essential products, such as food and beverages, cosmetics, and medicines. On the opposite, there was a significant drop in demand for non-essential products, such as fashion garments and accessories. For example, during the first two weeks of March 2020, H&M and Zara reported a 24.1% drop in sales [9]. In order to deal with this drop in demand, international firms such as Burberry, Gucci, and Prada re-purpose their production lines trying not to stop their factories, and started to produce face masks, gloves, and nonsurgical gowns [10].

Italy is one of the countries which was most affected by the early stages of the pandemic. A prolonged lockdown (9 March 2020–18 May 2020) was carried out to deal with the health emergency [11]. A particular important sector in the Italian economy is the footwear industry [12]. The fashion market is divided into different price ranges, and, except for the luxury sector, the most relevant segment is that of small and medium-sized enterprises (SMEs). In Italy, fashion companies have turned to production districts made up of SMEs where quality and craftsmanship represent the strengths [13]. However, Italian SMEs present limited resources and lack in terms of cognitive and organizational assets [14].

This paper aims to provide a case study related to two SMEs of the Italian footwear supply chain comparing sales and production data from pre-pandemic years with those affected by the COVID-19 pandemic. In particular, data are referred to two Tuscan SMEs. After conducting a descriptive analysis of the data, possible resilience factors of SMEs' fashion supply chains have been extracted, and strategies that SMEs could adopt to better cope with the disruption caused by the pandemic have been suggested.

The remainder of the paper is organized as follows. In Section 2, relevant literature review is provided. It analyses manuscripts dealing with supply chain management in the luxury fashion industry. In addition, we focused the attention on supply chain resilience. Section 3 outlines the research objective and methodology. Section 4 presents the real case of two Italian footwear companies, describing the production context under analysis, and the trend of the available data. Then, an analysis of the data and a discussion to highlight the effects of the pandemic on the supply chains is conducted in Section 5. In the same section, particular attention has been focused on potential resilience strategies that could be adopted to deal with a disruption, such as that caused by the COVID-19 pandemic. Finally, conclusions and possible future developments are reported in Section 6.

## 2. Literature Review

The main success factors of a fashion-luxury company are a high level of quality, the heritage of craftsmanship, the exclusivity of products, emotional appeal, brand reputation, recognizable style, and strong association with the country of origin. So far, the fashion-luxury sector has given greater importance to quality than production efficiency [15].

In Italy, fashion-luxury companies have turned to production districts made up of small and medium enterprises where quality and craftsmanship represent their strengths [13]. However, with the growth of fast fashion [3], luxury brands have also required suppliers to be more flexible and faster to meet market demand. This means rapidly changing production strategies, with the associated risk of encountering issues, during the implementation of improvements projects which involve the entire supply chain. Indeed, the management is particularly complex due to various factors that lead to the high unpredictability of the demand [15].

Usually, the design of a new collection of a fashion-luxury brand starts about six months before a fashion show's debut [16]. However, the market demand for new models is growing, so it is necessary to speed up the new product development processes [17]. According to [18], Choi et al. state that a key factor for the success of a new product is the time-to-market. New product development concerns several areas for improvements such as the development time of a new product and the easiness with which it can be industrialized [19]. Moreover, communication covers a fundamental role in this phase since new product development concerns multiple business areas and suppliers which are involved in the development and industrialization phases [20]. As stated in [21], many fashion brands move from the traditional original equipment manufacturing (OEM) strategy to the original design manufacturing (ODM) strategy. In this context, brands supervise and approve the progress of the work. Especially in the sampling and industrialization phase, revisions of the new models and exchanges of information are frequent to correctly conceive the final product. To improve collaboration between supply chain partners during the new product development phase, product lifecycle management (PLM) software is emerging. As stated in [22], these kinds of software reduce the product development time. In addition, even 'big data' can be used to help the design of new products helping in the choice of new articles to be developed [23].

In late 2019, the COVID-19 virus began to spread worldwide. The rapid evolution of the COVID-19 virus outbreaks led to a pandemic declaration by the World Health Organisation (WHO) on 11 March 2020 [24]. Due to its high transmissibility and serious health effects, the COVID-19 virus has led many countries to call for national lockdowns, causing multiple disruptions in supply chains.

The pandemic led to multiple changes in everyday life and within supply chains as well, such as the shortage of essential components [25], sudden variations in demand resulting in a re-purpose of the production [26], and workforce fluctuations related to the COVID-19 virus outbreak [27].

The COVID-19 pandemic has taken a heavy toll on the global fashion industry including some iconic fashion brands. Several governments close down manufacturing plants through store closures and event cancellations to slow the spread of the virus. From big fashion retailers to small apparel makers, all have been adversely impacted as reduced consumer spending on non-essentials resulted in decreased demand and margins and forced many brands to either shut shop or declare bankruptcy.

Since the beginning of the pandemic, the research community began to examine the impact of the COVID-19 pandemic on supply chains [8,28,29]. In particular, a topic of interest is the supply chain resilience, defined as " . . . the operational supply chain capability to withstand, adapt, and recover from disruptions at a minimal cost to ensure customer demand is fulfilled . . . " [30]. More than 200 papers have been published between 2020 and 2021 dealing with supply chain resilience strategies to withstand the pandemic.

In terms of mitigating supply chain disruption risk, studies have suggested that flexibility and diversification is the best way to hedge this risk [4]. By researching the domino effect of factors affecting supply chain resilience in the fashion industry, in [31], Bevilacqua et al. suggested that due to manufacturers highlighting flexibility in order fulfillment, a flexible production structure is vital to effectively address unpredictable turnarounds of the market on time. Likewise, in the research note about the COVID-19 pandemic and supply chain resilience, in [32], Ivanov and Das indicated that the focus of

supply chain resilience management should shift towards situational responses to real-time changes. In cases of unlikely but severe disruptions to supply chains, temporary sourcing diversification could prove to be an effective response strategy. Finally, in [33], Sreedevi and Saranga discussed how supply, manufacturing, and distribution flexibility moderate the relationship between environmental uncertainty and supply risk.

Based on this analysis, this paper presents a real case of two SMEs of the Italian footwear supply chain to highlight the main effects of the pandemic on the sector. Potential resilience factors of SMEs' fashion supply chains have been extracted, and strategies that SMEs could adopt to better cope with the disruption caused by the pandemic have been suggested.

## 3. Research Objective and Methodology

The main research questions of this paper are:

RQ1. What were the main effects of the pandemic on the SMEs of the Italian footwear sector?

RQ2. How did the Italian footwear SMEs react to the disruption caused by the COVID-19 pandemic, and which resilience strategies could they have adopted?

These research questions are addressed by discussing the real case of two Italian footwear companies. According to several authors, case research is the most suitable method for developing, testing, disproof, and/or refining a theory or hypothesis [34] as well as for the determination of further research needs [35], especially in a dynamic manufacturing system.

Case research was applied in order to study a phenomenon over time within its natural setting in two different sites. It has been employed in a positivist manner for theory building with the aim to evaluate how companies reacted to the disruption caused by the COVID-19 pandemic in order to focus on potential resilience strategies that could be adopted to deal with a disruption, such as that caused by the COVID-19 pandemic.

Our case research was performed using the following steps:

1. Analyzing and collecting information on the production flows and work cycles. Specifically, a detailed process analysis where the researchers analyzed the status of the management of orders on the entire supply chain was conducted;
2. Extracting and examining data from the engineering/production and supply chain management (SCM) modules of the enterprise resource planning (ERP) of the shoe factories;
3. Discussing the results obtained with the company managers by deepening the analyses conducted.

The results are reported in Section 4 using a real case study, while discussion and assessment on the level of resilience of the supply chain are reported in Section 5 with some suggestions of potential resilience improvement strategies. As such, the results and discussion section can be considered as a guideline for future practitioners.

## 4. Case Study

This section presents the real case of two Italian footwear companies, describing the production context under analysis and the trend of the available data.

### 4.1. General Description

Two Italian companies in the world of fashion footwear sector have been assisted in the analysis of their supply chains. In order to understand the dynamics of the supply chains, the standard production processes were initially studied and mapped, analyzing in detail the planning, programming, and control phase. In addition, interviews and informal brainstorming were conducted with the company's project team of experienced managers, planners, and operators to receive feedback on the data analyzed.

The companies produce high-quality footwear and production is characterized by a high level of craftsmanship and high production lead times, justified by a refinement of materials and a high level of precision in processing. Specifically, the following main peculiarities were detected:

- The presence of two sales seasons: autumn/winter (AW) and spring/summer (SS). The AW production season starts in March with the collection of customers' orders and ends in August (production and delivery phase starts around May), while SS production season starts in September with the collection of customers' orders and ends in February (production and delivery phase starts around November);
- The existence of a high level of obsolescence, which leads to the creation of new models every season. Short life cycles and high unpredictability of demand are usually due to the fact that the models on sale try to capture the trends of the moment, which are highly variable and driven by phenomena such as cinema and social media;
- A make-to-order (MTO) environment that led to the absence of materials in stock (e.g., raw materials) and production lead times of more than one month;
- Suppliers and subcontractors were located in Italy, while customers were located all over the world (e.g., Italy, Australia, Japan, USA).

In this context, orders from end retailers are collected at the beginning of the season to give manufacturers the time to organize themselves and place orders for raw materials and components. Usually, the seasonal production cycle of shoe factories consists of the following activities, which are sketched in Figure 1:

1. Collection of market data and new trends;
2. Creation of new models for the new season;
3. Gathering orders for reconducted products (models already present in old seasons) and new models;
4. Sending orders to suppliers and subcontractors;
5. Receipt of components;
6. Assembly of shoes;
7. Delivery to the end customer.

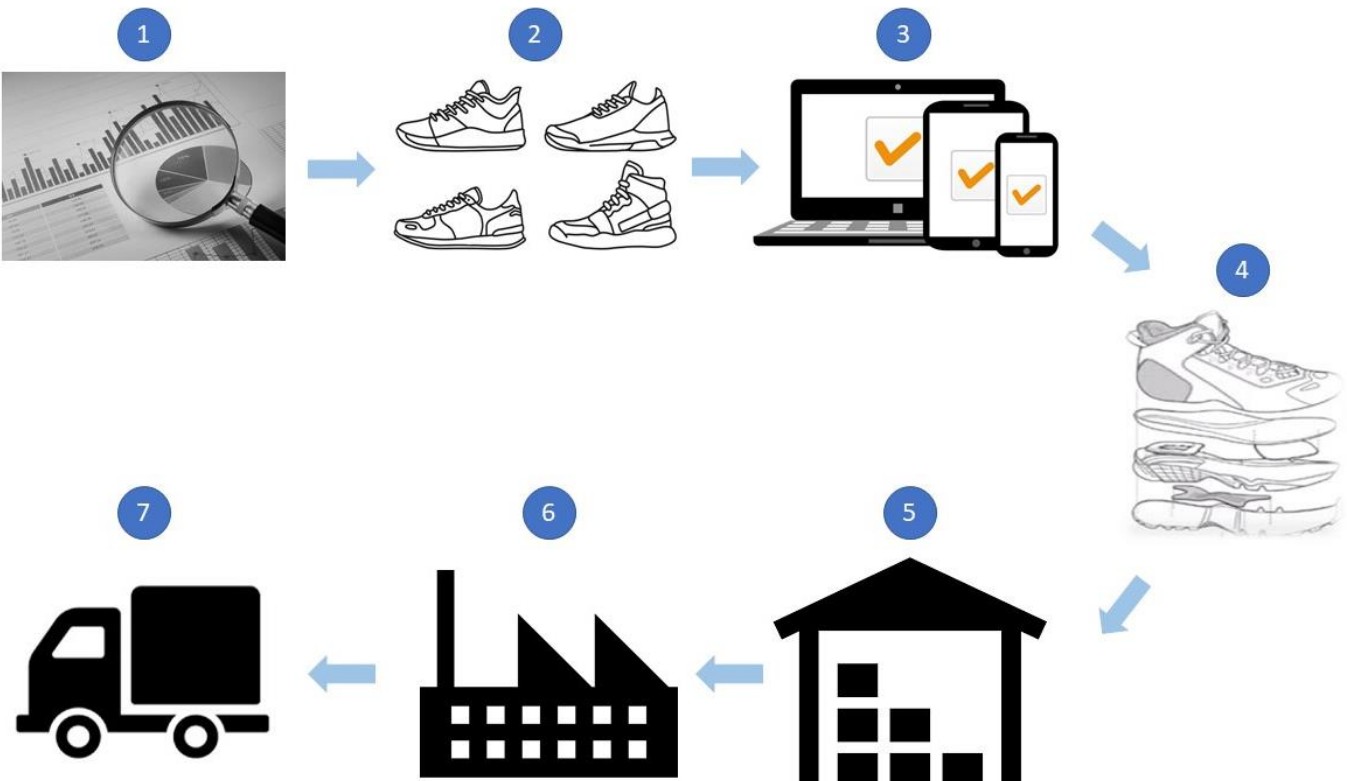

**Figure 1.** Shoe factory seasonal production cycle.

*4.2. Data Analysis*

According to the seasonal production cycle described in the previous subsection (Section 4.1), the supplier lead time, which can be defined as the amount of time that normally elapses between the time an order is received by a supplier and the time the order is shipped, was the most relevant. The companies provided data on their production phases or performed by subcontractors. On the contrary, suppliers did not provide data, but only estimated supply times collected during interviews as follows:

- Three to four weeks for tanneries, soles, and insoles suppliers;
- Two to three weeks for heels, uppers, and accessories suppliers.

In order to evaluate the impact of the COVID-19 pandemic, different production seasons were compared, comparing sales and production data from pre-pandemic years with those affected by the COVID-19 pandemic. Specifically, the following seasons were compared:

- Pre-pandemic seasons: AW 2019 season and SS 2020 season (from April 2019 to March 2020);
- Pandemic and post-pandemic seasons: AW 2020 season and SS 2021 season (from April 2020 to March 2021).

Inconsistent data (e.g., the end date of a production phase after the dating of shipment) and incomplete data (e.g., date of shipment missing) were removed. The available data were about 18,400 orders for shoe factory 1 (SF1) and about 8300 orders for shoe factory 2 (SF2).

Several key performance indicators (KPIs) and key parameters were assessed on both a seasonal and monthly basis. Orders had the production season associated with them, but to associate them with a specific month, the production end date of the order was taken as a reference. The following KPIs and parameters were taken into account:

- N° of pairs (the number of pairs of shoes produced/sold);
- N° of orders (the number of customers' orders received);
- Service Level (the ratio between the number of orders delivered on time, i.e., within the customers' due dates, and the total number of orders);
- Average production lead time (the mean time between the beginning of the first production stage and the final assembly of a shoe). This KPI considers the first production stage performed by the company or by a subcontractor (e.g., cutting of the leather). No data referred on suppliers were considered.

The next subsection (Section 4.2.1) maps the graphs and shows the trends of the KPIs.

4.2.1. Graphs and KPIs Trends

Figure 2 shows the seasonal trend of orders, the number of pairs produced, and the average lead time of SF1 (Figure 2a) and SF2 (Figure 2b).

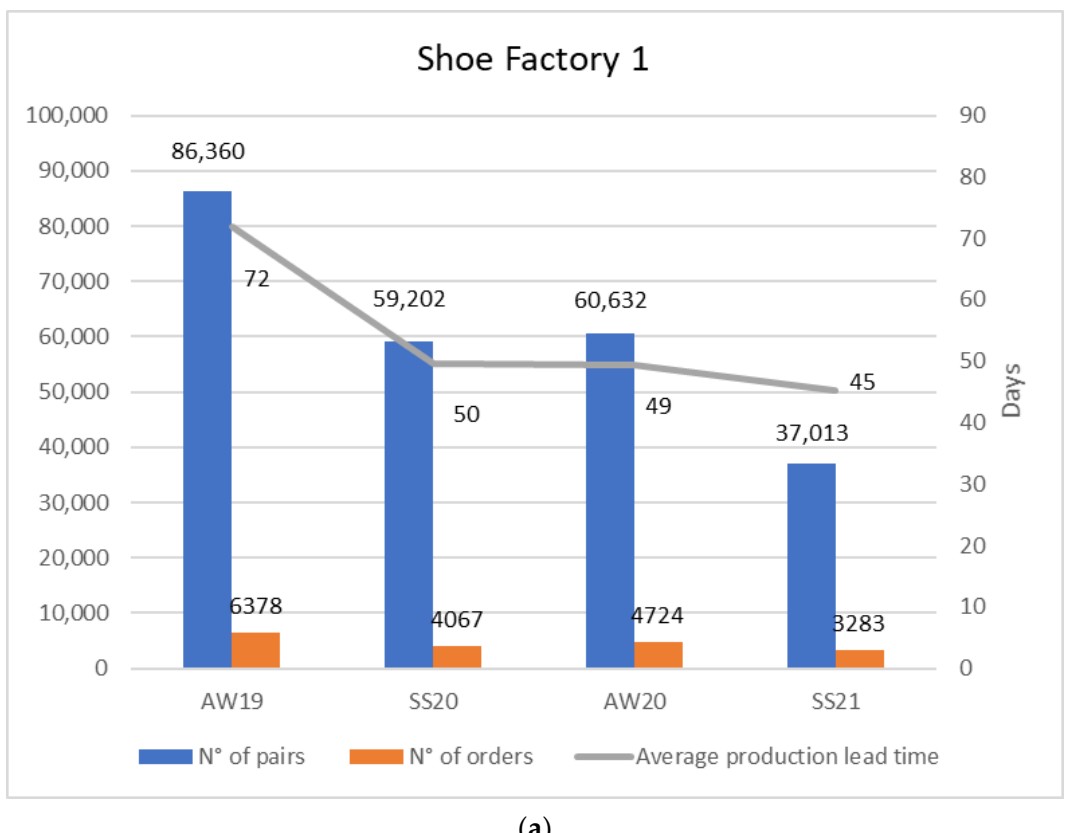

(**a**)

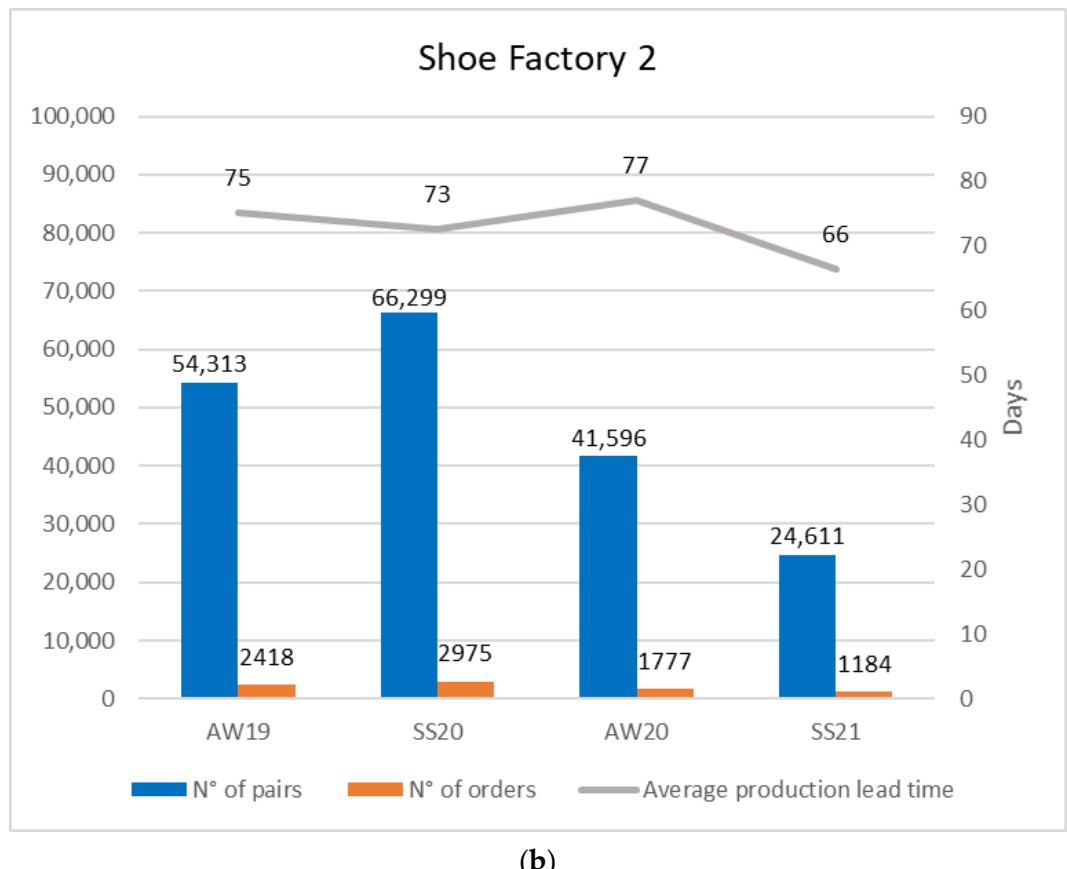

(**b**)

**Figure 2.** Shoe factories—seasonal N° of pairs, N° of orders, and average production lead time of SF1 (**a**) and SF2 (**b**).

Figure 3 shows the monthly trend of orders, and the average production lead time of SF1, considering the AW19-SS20 seasons (Figure 3a) and AW20-SS21 seasons (Figure 3b).

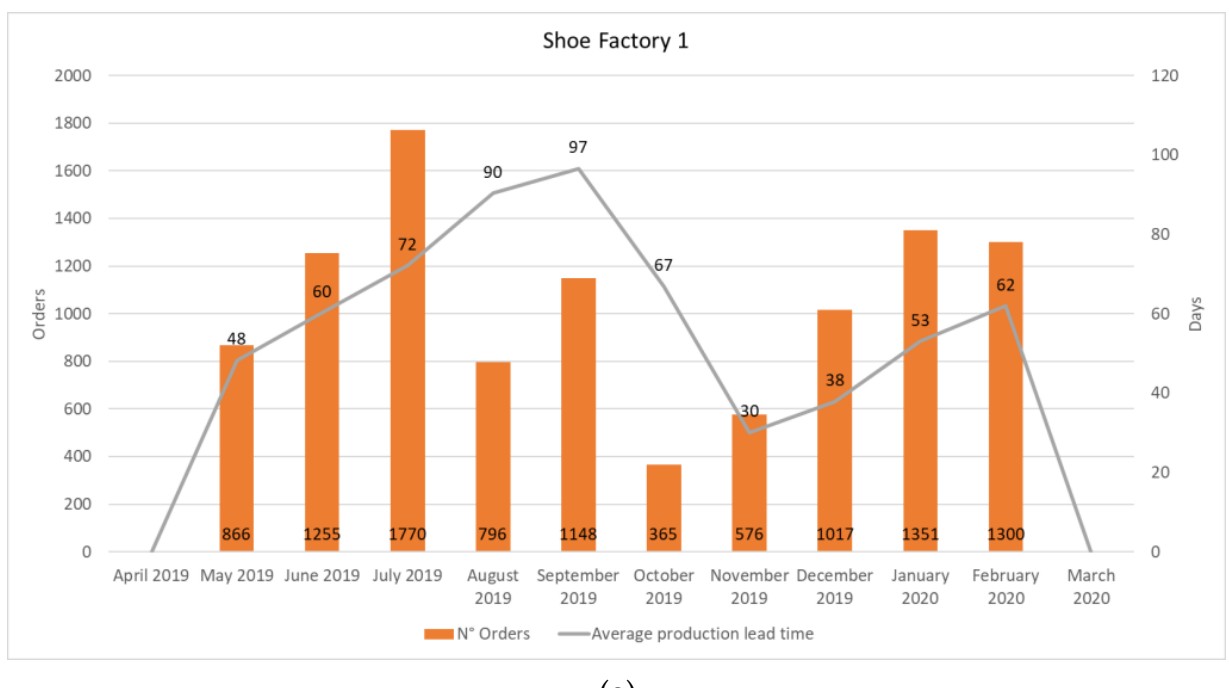

(**a**)

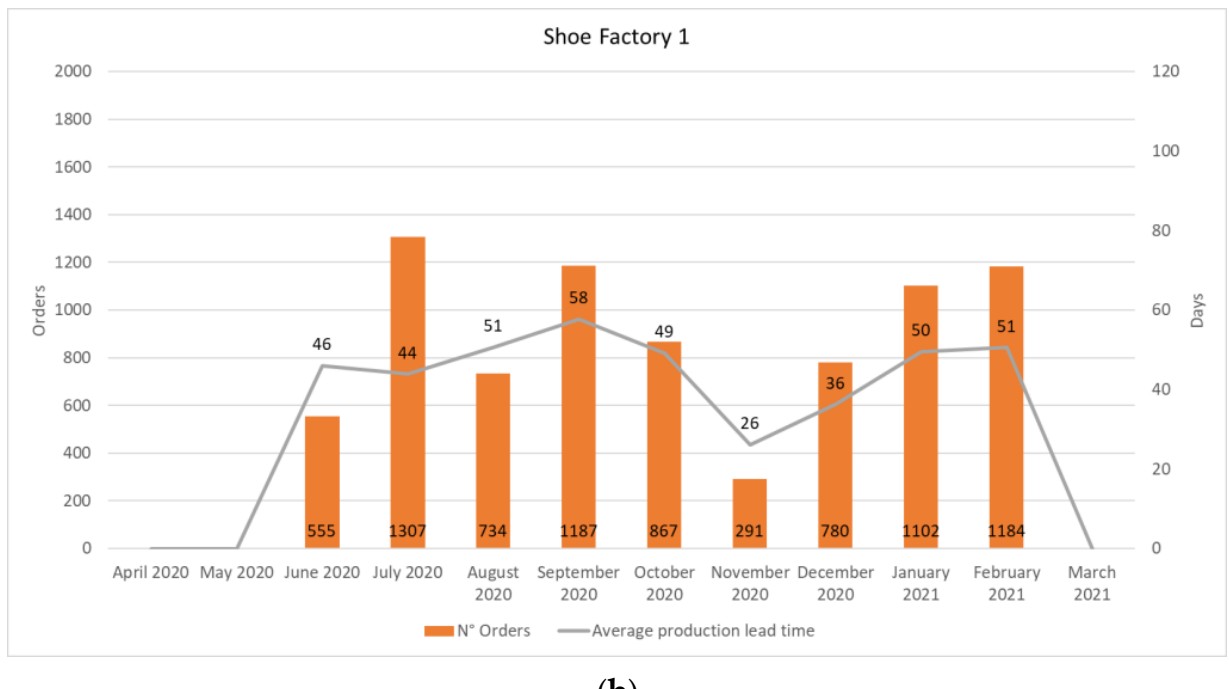

(**b**)

**Figure 3.** Shoe factories—seasonal N° of pairs, N° of orders, and average production lead time, considering AW19-SS20 seasons (**a**) and AW20-SS21 seasons (**b**).

Figure 4 shows the monthly trend of orders, and the average production lead time of SF2, considering the AW19-SS20 seasons (Figure 4a) and AW20-SS21 seasons (Figure 4b).

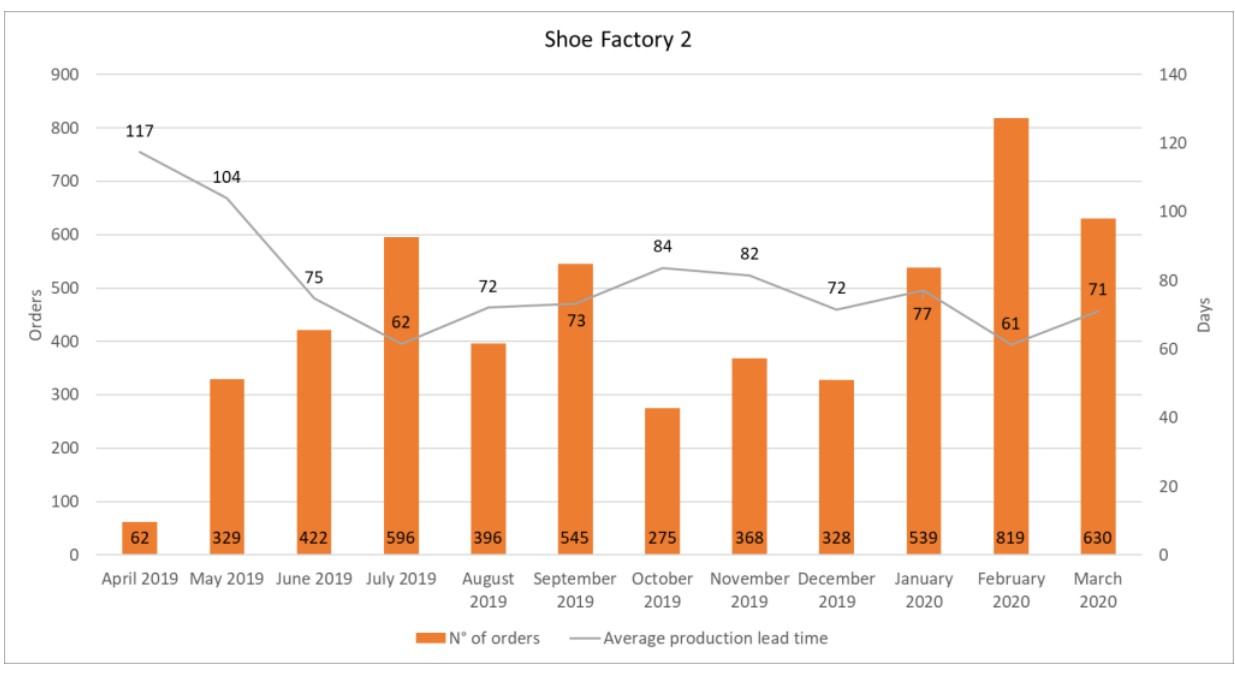

(**a**)

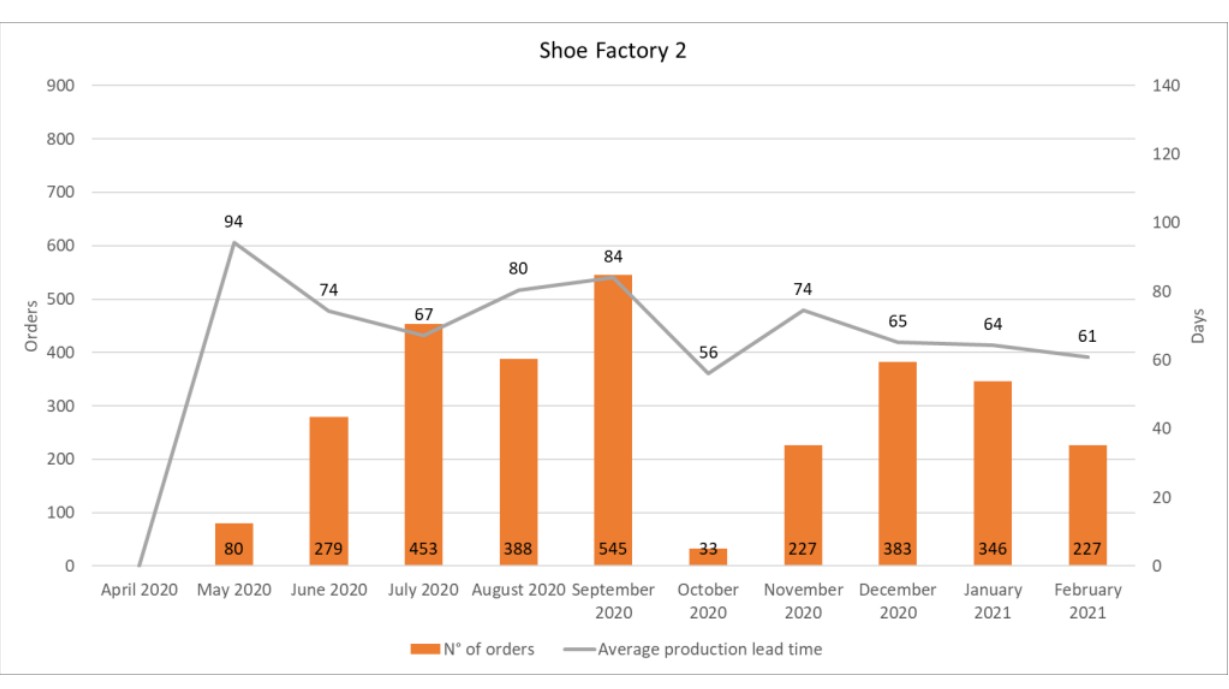

(**b**)

**Figure 4.** Shoe factory 2—monthly average production lead time, and N° of orders, considering AW19-SS20 seasons (**a**) and AW20-SS21 seasons (**b**).

Figure 5 shows the seasonal trend of orders, the number of pairs produced, and the Service Level of SF1 (Figure 5a) and SF2 (Figure 5b).

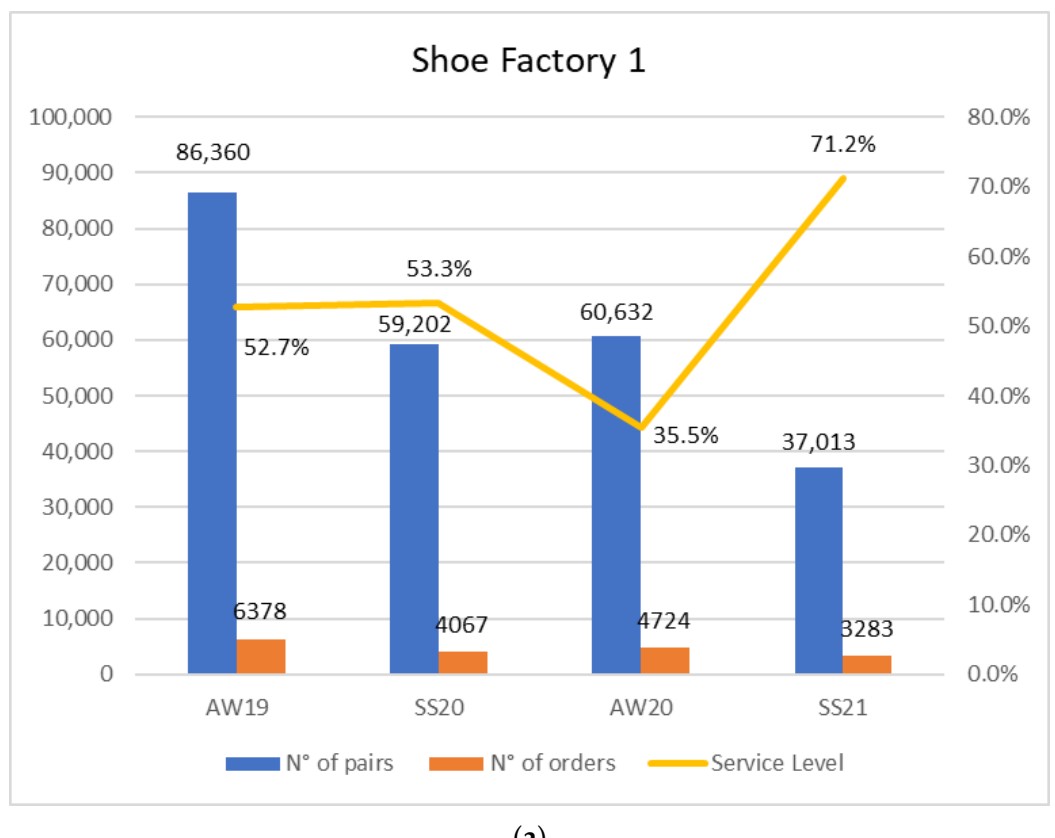

(**a**)

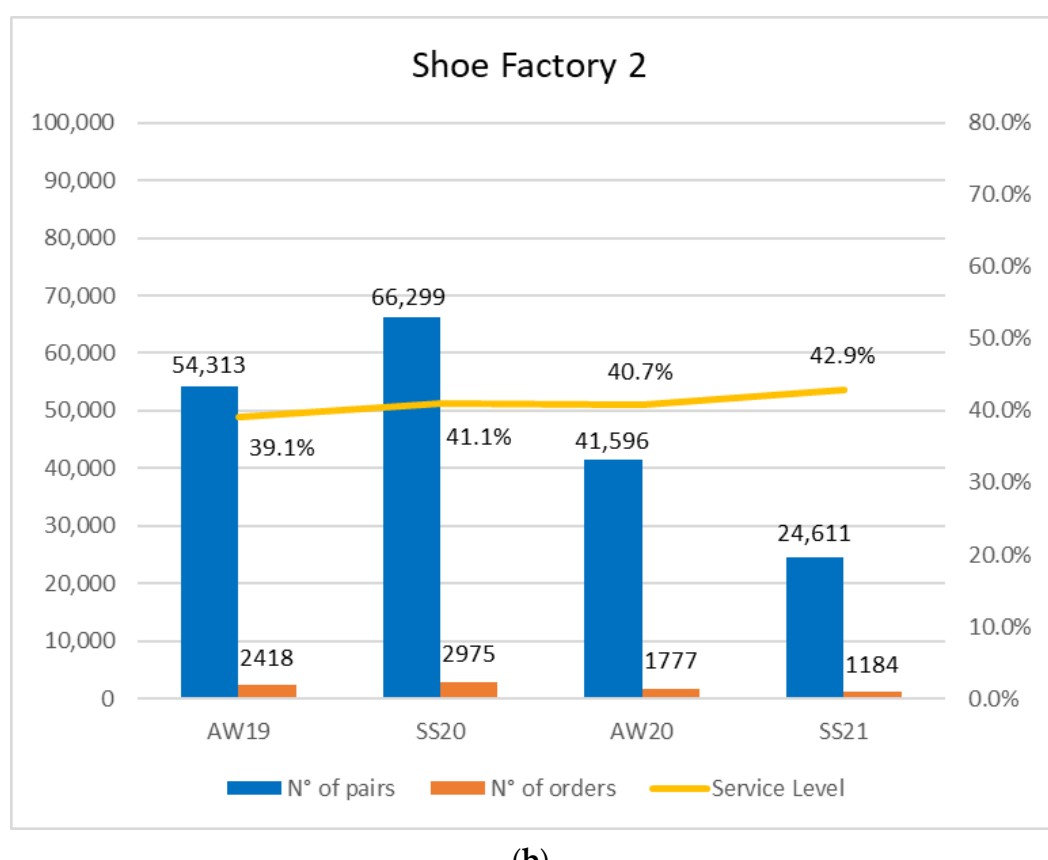

(**b**)

**Figure 5.** Shoe factories—seasonal N° of pairs, N° of orders, and Service Level of SF1 (**a**) and SF2 (**b**).

Figure 6 shows the monthly trend of orders, and the Service Level of SF1, considering AW19-SS20 seasons (Figure 6a) and AW20-SS21 seasons (Figure 6b).

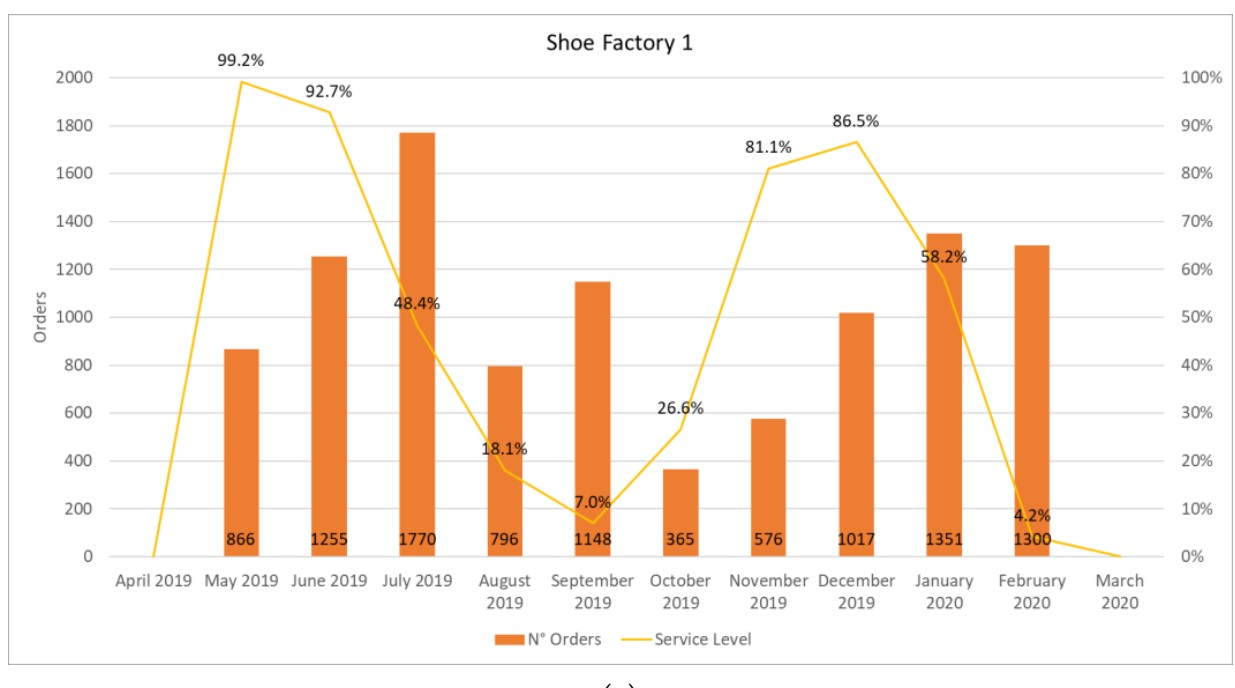

(**a**)

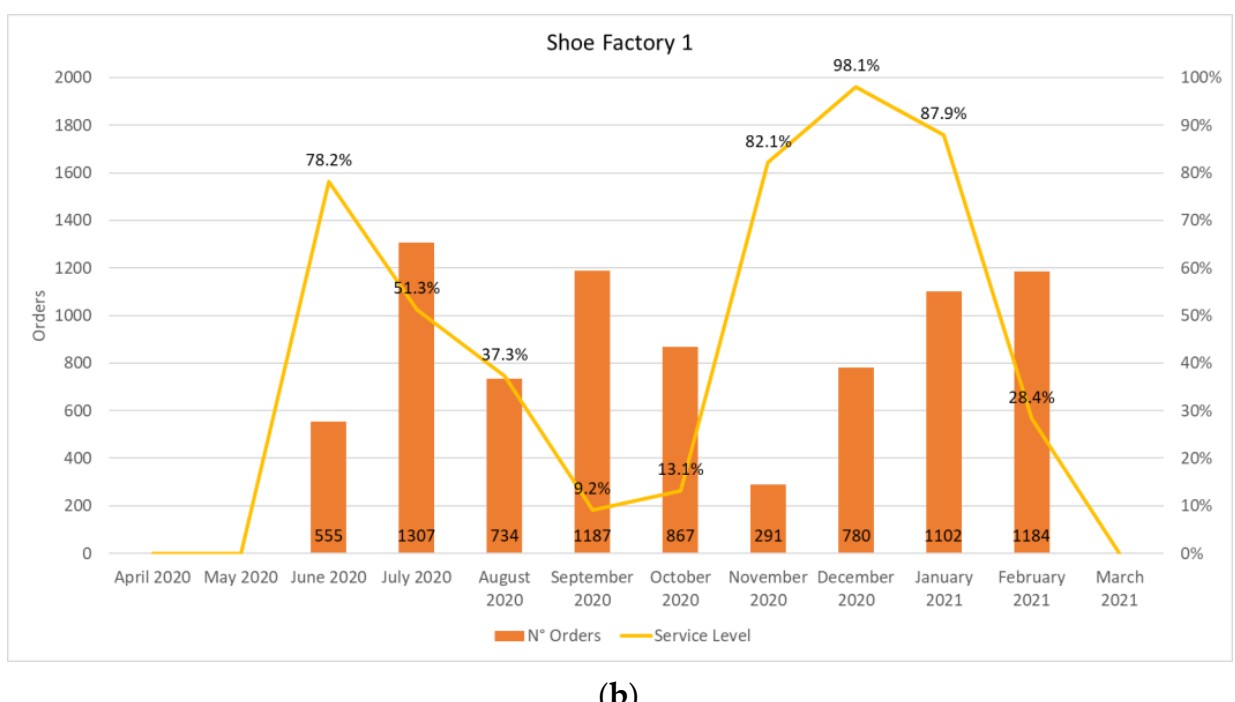

(**b**)

**Figure 6.** Shoe Factory 1—monthly service level and N° of orders, considering AW19-SS20 seasons (**a**) and AW20-SS21 seasons (**b**).

Figure 7 shows the monthly trend of orders, and the Service Level of SF2, considering AW19-SS20 seasons (Figure 7a) and AW20-SS21 seasons (Figure 7b).

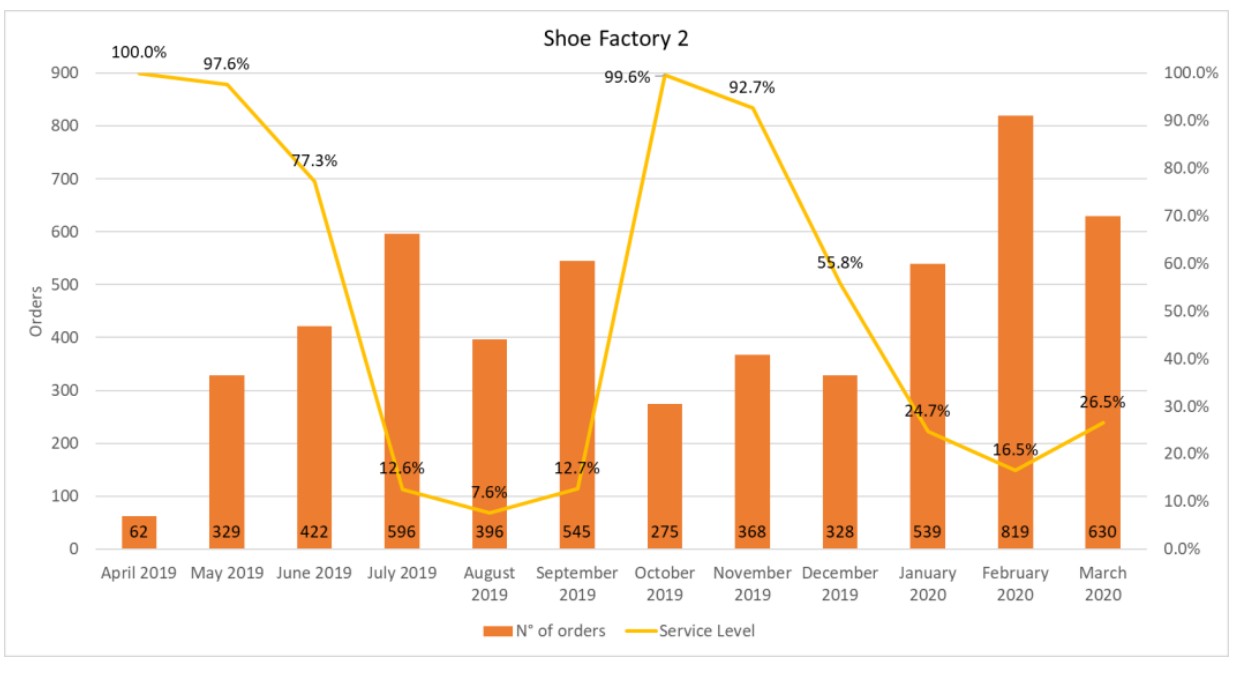

(**a**)

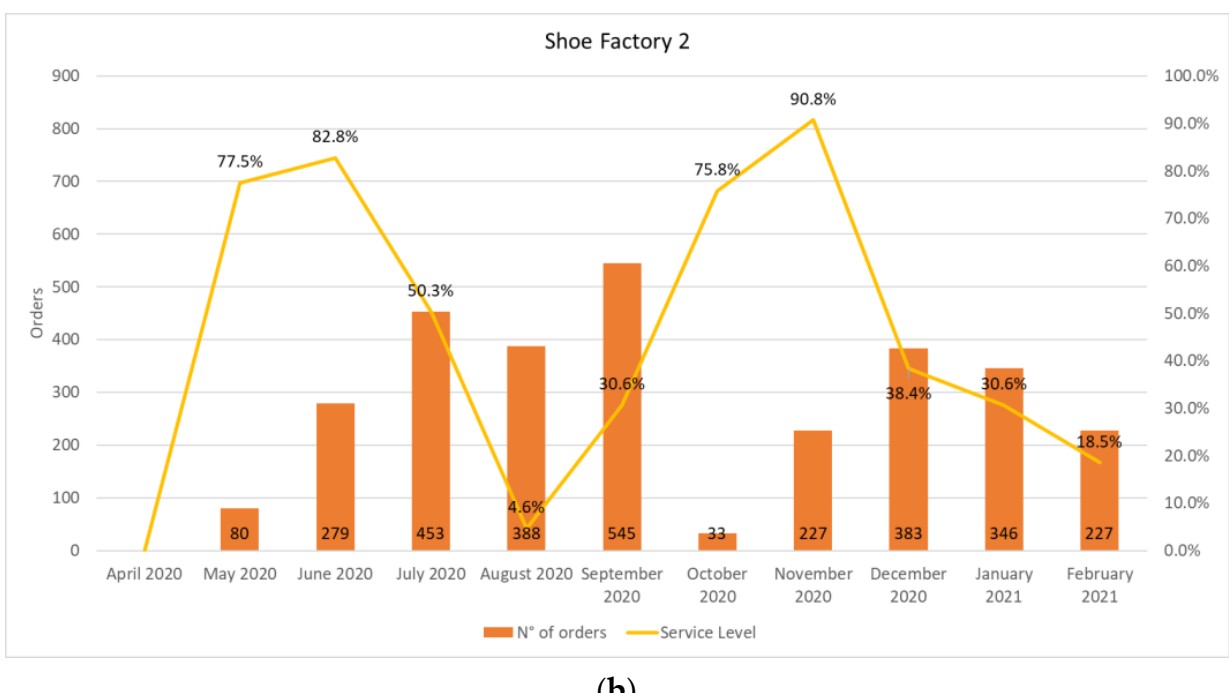

(**b**)

**Figure 7.** Shoe Factory 2—monthly Service Level and N° of orders, considering AW19-SS20 seasons (**a**) and AW20-SS21 seasons (**b**).

## 5. Discussion

The data available and plotted in Section 4.2.1 allow us to critically compare two production years: the first year, including the AW19-SS20 seasons unaffected by the pandemic, and the second year, including the AW20-SS21 seasons fully affected by the COVID-19 pandemic.

In the next subsection (Section 5.1), a particular type of engineering-based approach for measuring resilience that is based on the theoretical concept of the disaster resilience triangle is implemented. Then, seasonal trends (Section 5.2) and monthly trends (Section 5.3) will be

assessed to critically compare the performance of the companies in the pre-pandemic and post-pandemic seasons. Finally, evaluation on supply chain resilience has been reported in Section 5.4.

### 5.1. Measuring Resilience

From an engineering perspective, a common approach for assessing disaster resilience is to look at the physical characteristics of a system and to consider how system loss evolves by studying the extent to which the system is initially damaged and the amount of time needed for it to regain its normal functionality [36,37].

The discussion below builds on the work of Zobel [38,39], which focuses on characterizing the tradeoffs between the loss suffered by the system due to a disaster event and the subsequent system recovery time. Given a parameter, $T^*$, which is typically chosen to represent the maximum allowable recovery time for the process being modeled, and considering the average loss $\overline{X}$ over the duration of the disruption as the variable of interest, in Reference [40], Zobel defines predicted disaster resilience as follows:

$$R\left(\overline{X}, T\right) = \frac{T^* - \overline{X}}{T^*} = 1 - \frac{\overline{X}T}{T^*} \ \overline{X} \in [0, 1], \ T \in [0, T^*],$$

where $T$ is the time until recovery. By recovery, we considered the 10% around the monthly performance of the previous year.

Resilience values have been calculated for average production lead time ($R_{LT}$), N° of Orders ($R_{OR}$), and Service Level ($R_{SL}$). Table 1 shows the resilience values of the SFs.

**Table 1.** Resilience values of the SFs.

| Resilience Value | SF1 | SF2 |
|:---:|:---:|:---:|
| $R_{LT}$ | 0.84 | 0.83 |
| $R_{OR}$ | 0.69 | 0.61 |
| $R_{SL}$ | 0.81 | 0.80 |

This consistent interpretation allows us to compare the SFs in terms of the extent to which they were each impacted by the pandemic. The results thus show that all three dimensions are somewhat similar in their (normalized) average loss values, over the course of the pandemic, but different in the (normalized) length of time that they took them to recover. In particular, the time until recovery was about one month, three months and two months for SF1, and about two months, four months and two months for SF2 considering, respectively, lead time, N° of orders, and Service Level.

In the next subsections, seasonal trends (Section 5.2) and monthly trends (Section 5.3) will be assessed to critically compare the performance of the companies in the pre-pandemic and post-pandemic seasons.

### 5.2. Seasonal Trend Analysis

Starting with the analysis of graphs reported in Figure 2, it could be noted that a drop in sales has been recorded in the SS21 season. The AW20 season started during the COVID-19 pandemic, and the receipt of orders took place in March 2020, the first month of the Italian lockdown. However, this aspect did not affect the SF1 considering that the number of orders remained the same as the previous season (SS20). On the other hand, SF2 recorded a significant drop in sales in the AW20 season (−37% of pairs of shoes sold and −40% of orders compared to SS20; −23% of pairs of shoes sold and −27% of orders compared to AW19). As highlighted during the interviews, SF2 registered the cancellation of orders after the lockdown. However, in both cases there was a drop in sales in the SS21 season (percentages referred to the SS20 season):

- SF1: −37% in pairs of shoes sold and −20% in orders received;
- SF2: −63% in pairs of shoes sold and −60% in orders received;

The most significant drop was in pairs sold rather than in orders, although a 20% drop in orders is still significant. Continuing comparing the SS20 and SS21 and seasons and confirming the drop in the number of pairs of shoes sold, SF1 recorded a 22% drop in the average batch size of customers' orders (from about 15 pairs of shoes to 11 pairs of shoes) and SF2 a 10% drop in the average batch size of customers' orders (from about 23 pairs of shoes to 20 pairs of shoes).

Keeping the focus on Figure 2, it is possible to note that the production lead time has remained stable, with a slight improvement in the SS21 season. Compared to SS20, the last season without the impact of the COVID-19 pandemic, SF1 and SF2 recorded a 10% decrease in production lead times in the SS21 season.

Another relevant analysis that could be conducted comparing the production seasons is based on Service Level (Figure 5). SF1 recorded a negative peak in the AW20 season and then a clear improvement in the SS21 season. The managers of SF1 said that some due dates had been retracted with customers following lockdowns, but they did not update the ERP with the new due date. Consequently, the results of the analysis on the AW20 season may not be accurate. The average production lead time was stable, but all of the production activities had a one-month forward delay as a consequence of the two-month lockdown imposed by the Italian government). On the other hand, the positive peak registered during the SS21 season was associated with a decrease in orders and the ability to manage the workload received more easily. In the past, they accepted orders with delivery due dates that were often difficult to observe due to the workload already received.

Instead, the data are more reliable for SF2. The customers' due dates have been updated on the ERP and, in line with the stable production lead time, Service Levels remained constant, recording a slight improvement during the SS21 season.

*5.3. Monthly Trend Analysis*

The monthly analysis confirms the main finding of the seasonal analysis. Both SFs for the AW season began delivering shoes a month later. SF1 started completing orders in June for the AW20 season and in May for the AW19 season, while SF2 started completing orders in May instead of April. As reported in Figure 3, it is possible to assert that SF1 maintained the one-month delay in production until the following season (November 2020), indeed the minimum delivery of orders changed from October to November. On the other hand, SF2, excluding April and May 2020, maintained a trend such as the previous season (Figure 4).

Figures 3 and 4 highlight a drop in the number of orders received by both shoe factories, which recorded much lower order peaks. Table 2 resumes a quick comparison of the order peaks processed in one month comparing the respective seasons.

**Table 2.** Comparison between seasonal order peaks pre and post COVID-19 pandemic.

| Selling Season | | Autumn/Winter | | Spring/Summer | |
|---|---|---|---|---|---|
| Year | | 19 | 20 | 20 | 21 |
| Order peak | SF1 | 1770 | 1307 (−26.2%) | 1351 | 1184 (−12.4%) |
| | SF2 | 596 | 545 (−8.6%) | 819 | 383 (−53.2%) |

Seasonal production lead times trends are confirmed in the monthly analysis. Production lead times were stable with slight improvements in the SS21 season compared to SS20.

Finally, regarding Service Levels, SF1 experienced a gradual drop moving forward the AW20 season (Figure 6). The company management stated that after trying to meet the customers' due dates of the most relevant orders in June 2020 and not updating the ERP with the re-planned due dates, there was a gradual decline in the evaluated Service Level. Instead, the SS1 season showed improvements compared to the previous year. As mentioned in Section 5.2, the improvements were justified by the lower number of orders and the possibility to manage the workload more efficiently.

Regarding SF2, the Service Level remained stable, and also the trends recorded in the different years were similar (Figure 7). Positive peaks were recorded in the first months of seasonal production (May and June for AW seasons; October and November for SS seasons). Degradation of performance can then be observed in the following months of the seasons.

*5.4. Supply Chain Resilience Analysis*

During the COVID-19 pandemic, many supply chains experienced production slowdowns and were unprepared to deal with disruptions such as the ones caused by the COVID-19 pandemic [41]. Based on the analysis carried out, we wanted to extract the main effects that the COVID-19 pandemic caused to the analyzed SCs. In addition, suggestions to deal with future disruptions, in analogy to those caused by the COVID-19 pandemic will be provided.

Starting with the analysis of the performance of the SFs, they did not record any noticeable degradation. Excluding the months when they were unable to produce due to the Italian lockdown, they proved to be resilient and maintained stable their production performances. One resilience factor that proved to be crucial in dealing with the pandemic was the localization of suppliers. All suppliers had their closures synchronized with SFs and they did not cause any slowdowns. In addition, from a health point of view, both SFs stated that they had immediately implemented all of the new health protection standards to allow their employees to carry out their activities safely. This allowed the full resumption of the activities as soon as they received the authorization from the Italian government.

The COVID-19 pandemic has also led to a considerable drop in orders. The pandemic led to an economic crisis and changes in people's habits by limiting the purchase of non-essential goods such as footwear. To cope with this drop in sales, the footwear supply chain would have had to re-purpose their production lines and try to offer alternative products as carried out by brands in the same sector as Burberry, Gucci, and Prada [10]. However, the resources available for SMEs are not relevant, and changing the production lines and adapting them to different products may require significant investments. In such cases, external support would be needed, which could be found in the government. Alternatively, companies could collaborate to cope with emergencies such as, by a way of example, General Motors and Ventec, who formed a partnership to build ventilators for hospitals [42].

Innovation is at the heart of recovery for key players and fashion retailers as they consider new business models for survival. The recovery of growth in 2022 is expected to be largely driven by e-commerce, as consumers are still showing reluctance to make physical store visits, due to safety concerns as well as ongoing lockdowns. However, fashion retailing overall in many key regions is no longer defined by physical versus e-commerce, and the growing trend among apparel and footwear retailers and brands to offer a unified experience is becoming increasingly apparent.

**6. Conclusions**

This paper aims to provide a case study related to two SMEs of the Italian footwear supply chain comparing sales and production data from pre-pandemic years with those affected by the COVID-19 pandemic. In particular, data are referred to by two Italian SMEs. The case research method helped derive a richer, more contextualized, and more authentic interpretation of the phenomenon of interest than most other research methods by virtue of its ability to capture a rich array of contextual data.

After a brief description of the production sector under analysis, and identifying a set of relevant KPIs, the main performance variations recorded by the shoe factories under examination were highlighted. Based on the variations shown, the main resilience factors demonstrated by the Italian footwear supply chain were highlighted. Finally, possibilities of improvement from a supply chain resilience point of view were reported getting inspired by real scenarios, such as re-purposing or through partnerships.

Data were extracted from the engineering/production and supply chain management (SCM) modules of the ERP of the shoe factories. In the future, extracting and examining data from additional ERP modules (such as, by a way of example, human resources, sales and marketing, and purchase) could help to study the phenomenon of interest in a more detailed and contextualized manner. Furthermore, a questionnaire can be implemented as an additional data collection method to highlight additional factors related to the COVID-19 pandemic that were not taken into account.

In future studies, researchers might analyze other production sectors to extract relevant resilience strategies that can be adapted to deal with disruptions such as those caused by the COVID-19 pandemic. In addition, governments could encourage partnerships between different companies to create networks capable of dealing with disruptions such as the one caused by the COVID-19 pandemic.

**Author Contributions:** Conceptualization, M.B., L.M. and L.P.; methodology, M.B., L.M. and L.P.; validation, L.M. and L.P.; investigation, L.P.; writing—review and editing, M.B., L.M. and L.P.; supervision, M.B. All authors have read and agreed to the published version of the manuscript.

**Funding:** This research received no external funding.

**Institutional Review Board Statement:** Not applicable.

**Informed Consent Statement:** Not applicable.

**Data Availability Statement:** Not applicable.

**Acknowledgments:** This work was supported by Regione Toscana (POR FESR 2014-2020-Line 1.1.5.a3) through the Project QRLMS under CUP 4421.02102014.072000078.

**Conflicts of Interest:** The authors declare no conflict of interest.

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
