# Peer review of "The Impact of COVID-19 on the Italian Footwear Supply Chain of Small and Medium-Sized Enterprises (SMEs)—Evaluation of Two Case Studies"

_designs, 2022_

Round 1

Reviewer 1 Report

add theoretical studies based on relevant research results

Author Response

We really appreciated the comments from the Reviewers, that we wish to thank for their helpful comments that assured to improve our paper. We would also like to thank the Editor who gave us the opportunity to resubmit a revised version of the paper.

In the marked revised version of the paper, the text that we removed is highlighted with red colour and strikethrough font. The added and amended parts are marked with blue colour. When it is useful for the revision, we mentioned the sections of the marked revised paper that we modified in response to each Reviewers’ comment.

Finally, we made an ulterior linguistic revision of the paper.

All comments were carefully considered in this revision and we hope that the paper has reached the journal’s requirements.

Reviewer 2 Report

This is an interesting read to learn about the supply chain of luxury footware industry operating in Italy. However, before the recommendation for publication, the article has several limitations to overcome which are highlighted in the attached file. 

Author Response

(The authors gave the same response as above.)

Reviewer 3 Report

The article presents an interesting and up-dated analysis on the supply chain disruption on the companies' global performance.

The methodology appears relevant and the results similarly. 

Anyhow, certain improvements may be considered, at least for future research.

When considering KPI, the targets and comparison with other competitors may be relevant. The investigated factors are just part of potential sources for improving the companies' resilience. A broader analysis may be useful, but it's true, this is difficult to conduct as soon as companies are still in clear incertitude situations in terms of workload, human resource, supply chain, materials and accessories aso.

I like the approach and I think comparison between different companies and in the next step, between industries to propose relevant strategies for improving resilience may be valuable ; but broader analysis of factors and envisaged targets should be taken into account.    

Author Response

(The authors gave the same response as above.)

Round 2

Reviewer 2 Report

Suggested changes have been made. The paper may be accepted for publication.